# DOLPHIN: A Challenging and Diverse Benchmark for Arabic NLG

El Moatez Billah Nagoudi[ξ,⋆]    AbdelRahim Elmadany[ξ,⋆]
Ahmed Oumar El-Shangiti[λ]    Muhammad Abdul-Mageed[ξ,λ,⋆]

[ξ] Deep Learning & Natural Language Processing Group, The University of British Columbia
[λ]Department of Natural Language Processing & Department of Machine Learning, MBZUAI
{moatez.nagoudi,a.elmadany,muhammad.mageed}@ubc.ca

## Abstract

We present *Dolphin*, a novel benchmark that addresses the need for a natural language generation (NLG) evaluation framework dedicated to the wide collection of Arabic languages and varieties. The proposed benchmark encompasses a broad range of 13 different NLG tasks, including dialogue generation, question answering, machine translation, summarization, among others. *Dolphin* comprises a substantial corpus of 40 diverse and representative public datasets across 50 test splits, carefully curated to reflect real-world scenarios and the linguistic richness of Arabic. It sets a new standard for evaluating the performance and generalization capabilities of Arabic and multilingual models, promising to enable researchers to push the boundaries of current methodologies. We provide an extensive analysis of Dolphin, highlighting its diversity and identifying gaps in current Arabic NLG research. We also offer a public leaderboard that is both interactive and modular and evaluate several models on our benchmark, allowing us to set strong baselines against which researchers can compare.[1]

## 1 Introduction

Natural language generation (NLG) systems attempt to produce coherent, contextually appropriate, and linguistically accurate human-like language. These systems have a wide range of applications in everyday life, including in recreation, education, health, etc. The recent rise of generative models has transformed these NLG systems, making them more relevant and engaging than before. Crucial to measuring the performance of NLG systems are high-quality benchmarks. In particular, they provide standardized frameworks for comparing and quantitatively assessing differ-

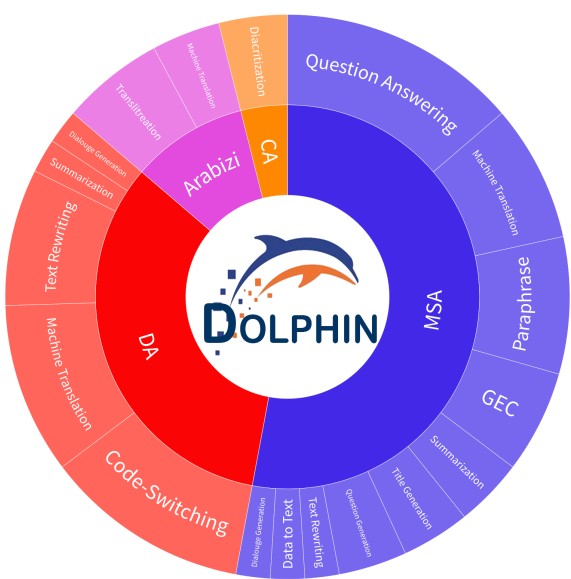

Figure 1: *Dolphin* task clusters and taxonomy. **GEC**: grammatical error correction. **CA**: Classical Arabic. **DA**: Dialectal Arabic. **MSA**: Modern Standard Arabic.

ent algorithms, models, and techniques. For NLG, benchmarks define specific criteria and metrics for evaluating performance, allowing for objectively gauging the strengths and limitations of different approaches and encouraging healthy competition. NLG benchmarks can also facilitate reproducibility and promote transparency across different studies, acting as a catalyst for advancement in the field.

Despite of this significance, efforts for developing nuanced NLG benchmarks that can allow us to track and guide performance on particular languages remain limited. For Arabic, a wide collection of languages and diverse varieties, there is currently no sizeable benchmark that caters to the needs of the community. In this work, we present a large benchmark for Arabic, dubbed *Dolphin*, to bridge this gap. Our novel benchmark is carefully curated to represent real-world usage of Arabic at scale. Dolphin covers *Classical Arabic (CA)*, a premodern standardized form of Arabic used for old

---

[1]https://dolphin.dlnlp.ai/.
⋆Equal contributions.

poetry and religious discourse that continues to be employed for literary expression and oration, *Modern Standard Arabic (MSA)*, a modern descendent of CA used in formal settings and in pan-Arab media, *dialectal Arabic (DA)*, such as varieties used in everyday communication in the different Arab countries. Dolphin also encompasses text written in both Arabic and Latin scripts, the latter usually referred to as *Arabizi*. The benchmark is comprised of 13 different generation tasks based on 40 different datasets across 50 test splits, making it by far the largest Arabic NLG benchmark to date and among the largest for any group of languages.

We build Dolphin on top of exclusively *public* datasets, adding a number of newly developed datasets of our creation. This makes Dolphin accessible and easy to use. Our benchmark is accompanied by a modular leaderboard with a unified evaluation metric, i.e., a *Dolphin score*. The leaderboard is designed to serve as a central hub for tracking and showcasing the performance of NLG systems. It functions as a dynamic and transparent platform where users can submit their models to compare their results against the state-of-the-art approaches. It also encourages a culture of transparency and detailed model description.

Overall, we make the following contributions: **(1)** We introduce a novel benchmark for Arabic NLG that is large, public, diverse, and inclusive. **(2)** We develop a dynamic leaderboard with a rich array of best design principles to facilitate the measurement of progress in the field. **(3)** We evaluate a wide host of Arabic and multilingual models on our benchmark, offering strong baselines. **(4)** We analyze our benchmark to identify gaps in existing work, hoping to help guide future directions. The rest of the paper is organized as follows: In Section 2, we provide an overview of related work. Section 3 introduces Dolphin design principles and task clusters. In Section 4, we present evaluations of the pretrained models on Dolphin, and discuss the results we acquire. We conclude in Section 5.

## 2 Related Works

Existing NLG benchmarks can be classified into three distinct categories: *Arabic-specific*, *X-specific* (where X refers to languages other than Arabic, such as English, Chinese, etc.), and *multilingual* benchmarks. In this section, we provide a brief overview of each category, highlighting their respective characteristics and scope. We offer more

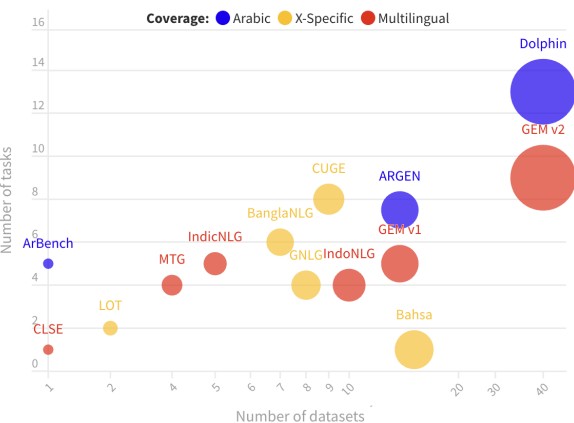

Figure 2: Comparison of the number of datasets and tasks supported by the Arabic (including *Dolphin*), X-specific, and Multilingual NLG benchmarks.

details on target languages, dataset sizes, and the breadth of tasks Dolphin covers in Appendix A. Table 1 and Figure 2 offer a summary of comparisons between Dolphin and other benchmarks.

**Arabic Benchmarks.** Sajjad et al. (2020) introduce AraBench, a machine translation (MT) evaluation benchmark consisting of five datasets for dialectal Arabic to English translation. AraOpus-20 (Nagoudi et al., 2022b) is another MT benchmark of parallel sentences between Arabic and 20 languages. Nagoudi et al. (2022a) introduce Ar-Gen, an Arabic NLG benchmark composed of 19 datasets covering seven tasks. In comparison, Dolphin is much larger, composed of exclusively *public* datasets, and covers more varieties. It is also the only benchmark accompanied by a leaderboard.

**X-Specific Benchmarks.** Liu et al. (2021) propose GLGE, a generation benchmark for English covering eight datasets across four tasks. CUGE (Yao et al., 2021) and LOT (Guan et al., 2022) are two Chinese benchmarks that cover both language understanding and generation tasks. BanglaNLG (Bhattacharjee et al., 2023) is a generation benchmark designed for Bangala comprising seven datasets across six tasks. Guntara et al. (2020) and Doan et al. (2021) present two MT benchmarks for Bahasa Indonesia and Vietnamese languages, respectively.

**Multi-Lingual NLG Benchmarks.** The generation evaluation and metrics benchmark (*GEM_v1*) (Gehrmann et al., 2021) is a multilingual benchmark environment for NLG. GEM_v1 features 18 languages across 13 datasets spanning five tasks. Gehrmann et al. (2022) propose a second version, GEM_v2, with a new set of datasets and

| Category | Benchmark | Reference | Task Clusters | Lang | Datasets | #Clusters |
|---|---|---|---|---|---|---|
| Arabic | DOLPHIN | *Our work* | *ADT, CS, DRG, DT, GES, MT, NTG, PPH, QA, QG, TRW, TRS, TS* | Ar | 40 | 13 |
| | ArBench | Sajjad et al. (2020) | *MT* | Ar | 5 | 1 |
| | AraOPUS-20 | Nagoudi et al. (2022b) | *MT* | Ar | 1 | 1 |
| | ARGEN | Nagoudi et al. (2022a) | *CS, MT, NTG, PPH, QG, TS, TRS* | Ar | 13 | 7 |
| X-Specific | GNLG | Liu et al. (2021) | *DRG, DT, TT, TS* | En | 8 | 4 |
| | BanglaNLG | Bhattacharjee et al. (2023) | *MT, TS, QA, DRG, NTG, CLTS* | Bn | 7 | 6 |
| | CUGE | Yao et al. (2021) | *QA, DR, TS, DT, DRG, MT, CLTS, MC* | Zh | 9 | 8 |
| | Bahasa Indonesia | Guntara et al. (2020) | *MT* | Id | 14 | 1 |
| | PhoMT | Doan et al. (2021) | *MT* | Vi | 1 | 1 |
| | LOT | Guan et al. (2022) | *RES, DS* | Zh | 2 | 2 |
| Multilingual | CLSE | Chuklin et al. (2022) | DRG | 3 | 1 | 1 |
| | GEM$_{v1}$ | Gehrmann et al. (2021) | *DRG, DT, RES, TS, SMP* | 18 | 13 | 5 |
| | GEM$_{v2}$ | Gehrmann et al. (2022) | *DRG, DT, PPH, QA, QG, RES, SLG, SMP, TS* | 51 | 40 | 9 |
| | IndicNLG | Kumar et al. (2022) | *NTG, TS, PPH, QG, BG* | 11 | 5 | 5 |
| | MTG | Chen et al. (2022) | *SG, QG, NTG, TS* | 5 | 4 | 4 |
| | IndoNLG | Cahyawijaya et al. (2021) | *TS, QA, CC, MT* | 3 | 10 | 4 |

Table 1: Comparison of NLG benchmarks proposed in the literature across the different covered task clusters. **ADT**: Arabic text diacritization. **CS**: Code-Switching. **DRG**: dialogue response generation. **DT**: data-to-text. **GEC**: grammatical error correction. **MT**: machine translation. **NTG**: news title generation. **PPH**: paraphrase. **QA**: question answering. **QG**: question generation. **RES**: reasoning. **SLG**: slide generation. **SMP**: text simplification. **TRS**: transliteration. **TRW**: text rewriting. **TS**: text summarization. **TT**: table to text. **CLTS**: cross-lingual text summarization. **MC**: math computation. **DR**: document retrieval. **DS**: discourse structure. **CC**: chit-chat. **BG**: biography generation. **SG**: story generation.

more challenging tasks. This new version supports 40 documented datasets in 51 languages. Other multilingual NLG benchmarks include CLSE (Chuklin et al., 2022), IndoNLG (Cahyawijaya et al., 2021), IndicNLG (Kumar et al., 2022), and MTG (Chen et al., 2022). As Figure 2 shows, compared to these benchmarks, Dolphin is the largest both in terms of the number of tasks and datasets. We now introduce Dolphin.

## 3 Dolphin Benchmark

Our objective is to provide a comprehensive and challenging benchmark for natural language generation that enables the assessment of language models and the tracking of progress in Arabic. To attain this objective, we develop Dolphin , considering several design principles that we will now elucidate.

### 3.1 Design Principles

**Wide, diverse coverage.** As our goal is to offer a demanding and diverse benchmark, we incorporate *as many datasets from as many tasks as is feasible*. This allows comprehensive evaluations of LMs. It also facilitates painting as complete a picture as possible of the limits of current methods across the different tasks. Reflecting this principle, our benchmark is large. It comprises 40 distinct datasets, covering 13 different task clusters.

**Public datasets.** A major principle in choosing our datasets is public accessibility as it enables researchers to train and evaluate models without incurring expenses associated with acquiring private data. For this reason, all our 40 datasets are publicly available.

**Rich linguistic variability.** In order to accurately reflect the multifaceted and diverse nature of Arabic languages and dialectal varieties, we strategically incorporate datasets collated from an array of sources, each corresponding to different sociological and orthographic traditions. Specifically, we construct Dolphin considering four major variants of Arabic: Arabizi (an unconventional method where Arabic is transcribed in Latin script); Classical Arabic (CA); Dialectal Arabic (DA) from a myriad of cities, countries, and regions; and Modern Standard Arabic (MSA). The heterogeneous nature of our datasets allows for a comprehensive representation of Arabic across wide linguistic nuances and orthographic traditions. Refer to Figure 1 for an illustrative depiction of the distribution of our datasets over various Arabic varieties for each specific task. Table 2 provides a quantitative description of these varieties in Dolphin.

**Standard evaluation metrics.** Most generation tasks can be evaluated using traditional automated metrics such as BLEU (Papineni et al., 2002) and ROUGE (Lin, 2004). Both of these metrics evaluate the n-gram overlap between a reference text

and the generated text. Nevertheless, in many tasks (e.g., question generation, open domain generation, title generation) there are multiple valid ways to produce a given text. In our benchmark, in addition to $F_1$, BLEU, and ROUGE, we use several other evaluation metrics such MaxMatch (M2) (Dahlmeier and Ng, 2012) for grammatical error correction, and Character Error Rate (CER) (Morris et al., 2004) for diacritization.

**Modular, interactive leaderboard.** To support future research, we develop a public leaderboard that enables the evaluation of multilingual and Arabic LMs on Dolphin. Our leaderboard is interactive and provides detailed metadata about the corpora such as size, training, development, and test splits, data sources (e.g., URL, GitHub), and citations to publications. The leaderboard also offers details of language models assessed such as the number of parameters, epochs to conversion, pretraining and finetuning information, etc. We provide a screenshot from our leaderboard in Figure D.1. We now introduce each of the task clusters in Dolphin.

| Task Variety | # Clusters | # Datasets | # Test Sets |
|---|---|---|---|
| *Arabizi → X* | 1 | 2 | 2 |
| *Arabizi → MSA* | 1 | 3 | 3 |
| *CA → CA* | 1 | 1 | 1 |
| *DA → DA* | 2 | 2 | 3 |
| *DA → MSA* | 1 | 1 | 4 |
| *DA → En* | 1 | 1 | 5 |
| *DA-X → X* | 1 | 1 | 6 |
| *Table → MSA* | 1 | 1 | 1 |
| *MSA → MSA* | 7 | 21 | 21 |
| *X → MSA* | 1 | 2 | 4 |

Table 2: Descriptive statistics of the linguistic diversity in Dolphin across the different data splits.

## 3.2 Task Clusters

Dolphin involves 50 test sets curated from 40 datasets. We arrange Dolphin into 13 task clusters, as follows: (1) machine translation, (2) code-switching, (3) text summarisation, (4) news title generation, (5) question answering, (6) question generation, (7) transliteration, (8) paraphrasing, (9) text rewriting, (10) diacritization, (11) data-to-text, (12) dialogue generation, and (13) grammatical error correction. Appendix Table B.2 shows a summary of the data splits across datasets and task clusters in Dolphin. We present each task cluster in Dolphin next.

### 3.2.1 Machine Translation

The MT cluster is built around three tasks: **(1) $X \rightarrow$ MSA.** In this task, we test the ability of the models to translate from six foreign languages into MSA. We use the UN parallel corpus (Ziemski et al., 2016), a dataset covering the six official UN languages (i.e., Arabic, Chinese, English, French, Russian, and Spanish). The UN corpus consists of development and test sets only.[2] For training, we randomly select 50K *X*-Arabic parallel sentences from the multilingual corpus MultiUN (Eisele and Chen, 2010) where *X* is a language from the six official languages.
**(2) Arabizi $\rightarrow$ X.** The goal of this task is to translate from Arabizi dialectal text[3] into one of two foreign languages French and English. For this, we use Darija (Outchakoucht and Es-Samaali, 2021) and NArabizi (Seddah et al., 2020).
**(3) Dialects $\rightarrow$ English.** For this task, we focus on MT from six Arabic dialects into English using the MDP corpus (Bouamor et al., 2014). MDP is a human-translated collection of 1K sentences in Egyptian, Tunisian, Jordanian, Palestinian, and Syrian Arabic, in addition to English. For training, we use the 10K MSA-English manually translated sentences proposed by Bouamor et al. (2018) under a 'zero-shot' condition.[4]

### 3.2.2 Code-Switching

The purpose of the code-switching (CS) task cluster is to translate Arabic dialect text that includes code-switching with a foreign language into that foreign language. For this, we create six new human-written (natural) code-switched parallel test datasets, under two tasks: **(1) DIA-FR $\rightarrow$ FR.** This consists of 300 code-switched Arabic-French tweets collected from Algerian, Moroccan, and Tunisian Twitter. **(2) DIA-EN $\rightarrow$ EN.** This is collected from Egyptian, Jordanian, and Palestinian Twitter and consists of 300 code-switched Arabic-English posts. For both of these DIA-FR and DIA-EN tasks, a human translation is performed by one native speaker from each dialect with semi-native English/French fluency. For these two tasks, we perform experiments under the zero-shot setting. That is, we use no actual *code-*

---

[2] 4K sentences that are aligned across all official languages.

[3] **Arabizi** is the romanization of Arabic script (Darwish, 2013). In this task, we investigate the Algerian and Moroccan Arabizi.

[4] Due to lexical overlap between Arabic dialects and MSA, this is not zero-shot in the strict sense of the word.

*switched training* data. Rather, we extract 50K MSA-English and MSA-French sentences from AraOPUS-20 (Nagoudi et al., 2022b) that we use for *monolingual training*. We then extract 50 pairs from each code-switched dialect pair for development and test on the 250 remainder sentences.

### 3.2.3 Text Summarization

For the text summarization (TS) cluster, we use the following five Arabic and multilingual (including Arabic) publicly available datasets: (1) MassiveSum (Varab and Schluter, 2021), (2) XL-Sum Hasan et al. (2021), (3) CrossSum (Bhattacharjee et al., 2021), (4) ANT (Chouigui et al., 2021), and (5) MarSum (Gaanoun et al., 2022).

### 3.2.4 News Title Generation

The news title generation (NTG) task is about producing a suitable title for a given news article. That is, a title generation model is required to output a short grammatical sequence of words that are appropriate for the content of the article. For this, we use two datasets: (1) Arabic NTG (Nagoudi et al., 2022a), and (2) XLSum (Hasan et al., 2021).[5]

### 3.2.5 Question Answering

For the QA cluster, we use seven publicly available QA datasets across four tasks. A summary of the QA cluster is in Appendix Table B.2. We also provide brief information about each task here.

**Extractive QA.** We use four publicly available QA datasets: (1) The Arabic QA dataset ARCD (Mozannar et al., 2019) and the Arabic part of the following three multi-lingual QA test sets: (2) MLQA (Lewis et al., 2019), (3) XQuAD (Artetxe et al., 2020), and (4) TyDiQA (Artetxe et al., 2020). For all the extractive QA experiments, we finetune on the GoldP multilingual TyDiQA$_{train}$ (Artetxe et al., 2020) and evaluate on the test sets listed above.

**Retrieval QA.** For this task, we use (5) LAReQA (Roy et al., 2020), a cross-lingual retrieval QA dataset built by converting the extractive QA dataset XQuAD (Artetxe et al., 2020) into a retrieval task XQuAD-R. In our benchmark, we focus on the Arabic part of XQuAD-R (AraQuAD-R).

**Open-Domain QA.** In this task, the goal is to answer fact-based questions in natural language. We

add (6) DAWQAS, an Arabic *Why* QA dataset (Ismail and Nabhan Homsi, 2018) to our QA cluster.

**Multi-choice QA.** We also use (7) EXAMS (Hardalov et al., 2020), a cross-lingual multi-choice QA dataset that covers 26 languages (including Arabic). Since we only have this particular test set for Arabic, we follow Hardalov et al. (2020) in evaluating the models on EXAMS under a zero-shot setting.[6]

### 3.2.6 Question Generation

The question generation (QG) cluster involves generating a question for a given passage (Gehrmann et al., 2021). The model is trained to generate simple questions relevant to passages along with their answers. For this cluster, we use (passage, answer, and question) triplets from five out of the seven QA question datasets described in Section 3.2.5.[7]

### 3.2.7 Paraphrase

The main goal of this task is to produce for a given Arabic sentence a paraphrase with the same meaning. For this, we employ the following four datasets: (1) AraPara, a multi-domain Arabic paraphrase dataset (Nagoudi et al., 2022a), (2) ASEP, an Arabic SemEval paraphrasing dataset (Cer et al., 2017), (3) Arabic paraphrasing benchmark (APB) (Alian et al., 2019), and (4) the Arabic section of TaPaCo (Scherrer, 2020), a multilingual paraphrase corpus.

### 3.2.8 Transliteration

The task of transliteration (TS) is about converting a word or text from one writing system to another while preserving the pronunciation and sound of the original language. We create our TS component using three word-level datasets, as follows: **(1) ANETA**, an English-Arabic named entity transliteration and classification dataset proposed by Ameur et al. (2019). **(2) ATAR** (Talafha et al., 2021), a word-level parallel corpus containing human translations between Jordanian Arabizi[8] and MSA. **(3) NETransliteration** (Merhav and Ash, 2018), a bilingual named entity (person names) transliteration dataset mined from Wikidata for English to each of Arabic, Hebrew, Japanese, Katakana, and Russian.

---

[5]We note that XLSum (Hasan et al., 2021) has news articles annotated with summaries and titles. We use pairs of articles and titles to create the title generation data.

[6]we use the multilingual part for Train and Dev, where no Arabic data is included, and blind-test on the Arabic test split.

[7]We exclude the multi-choice QA EXAMS (Hardalov et al., 2020), the open-domain QA DAWQAS (Ismail and Nabhan Homsi, 2018).

[8]An informal variant of Arabic spoken in Jordan

| Dataset | Source | Target |
|---|---|---|
| ADT | ... ولا أنهى الكلام على الطهارة المائية صغرى وكبرى انتقل يتكلم على الطهارة الترابية | ... وَلَمَّا أَنْهَى الْكَلَامَ عَلَى الطَّهَارَةِ الْمَائِيَّةِ صُغْرَى وَكُبْرَى انْتَقَلَ يَتَكَلَّمُ عَلَى الطَّهَارَةِ التُّرَابِيَّةِ |
| AEC | لم أصدق أن صديقي بدأ العمل معي | أين تعملون يا رفاق؟ |
| DRG$_{EGY}$ | باين عليك قلقان. خير حاصل معاك حاجة؟ | والله تعبان جدا من ظروف البلد الصعبة |
| DRG$_{GUL}$ | ما يعجبني أنو أخوي يروح هنا وهناك بدون ما يسوي أي شي مفيد. | ممكن هو يحاول يدور على شغل. هينحل كل شي لن يلقى واحد. |
| DRG$_{LEV}$ | ما تنسى تفرشي سنانك قبل ما تنام | ايه هياني قايم |
| TAPACO | هي قدمت لي بعض الطعام. | هي ذودتني ببعض الطعام. |
| AraPara | لا يسري هذا العرض بالاقتران مع عروض أخرى . | العرض غير صالح مع أي ترقية أخرى. |
| APB | سمعت صوت الأسد . | سمعت زئير الأسد . |
| APGC | نوم العوافي ، هل ترغين من أمك أن تبقى ؟ | نوم العوافي ، هل ترغب من أمك أن تبقى ؟ |
| MT Darija | mzyana had jrida fach tatkon zwina b7al haka | It's nice, this park, when it looks good like that |
| MT EN → MSA | the report is silent about temporary special measures. | لم يرد في التقرير أي ذكر للتدابير الخاصة المؤقتة. |
| CS JO → EN | منيح coating أو يعمله alloy يا يغير ال | Either he changed the alloy or make it a good coating |

Table 3: Examples from datasets included in *Dolphin* .

### 3.2.9 Text Rewriting

The text rewriting (TR) cluster is about generating a text of the target style while preserving the content of the source input text. The TR cluster contains two tasks: **(1) DIA → MSA.** This task involves converting a text written in an Arabic dialect into MSA. For this, we use Dial2MSA (Mubarak, 2018). Dial2MSA is a parallel dialectal Arabic corpus for converting Egyptian, Maghrebi, Levantine, and Gulf dialects into MSA. **(2) Gender Rewriting.** We use the Arabic parallel gender corpus (APGC) proposed by Alhafni et al. (2022), where the task is to take a given input sentence written in one gender (e.g., male) to produce a target sentence that has the same meaning but employing the opposite gender (i.e., female).

### 3.2.10 Diacritization

Arabic text diacritization (ATD) is the computational process of restoring missing diacritics or vowels to the orthographic word or a sequence of words (i.e., a sentence or a whole text). For this task, we use the Arabic diacritization dataset proposed by Fadel et al. (2019).

### 3.2.11 Dialogue Response Generation

Dialogue response generation (DRG) is a human-computer interaction task with the goal of automatically producing a human-like response given a dialogue context. In this cluster, we have two tasks: **(1) MSA DRG.** For this task, we use the Arabic empathetic chatbot (AEC) dataset (Naous et al., 2020). It contains open-domain utterances with their corresponding empathetic responses machine translated from English into MSA. **(2) Dialectal DRG.** We add the open-domain response generation in Arabic dialects proposed by Naous et al. (2023). Three native translators from the Levantine,

Egyptian, and Gulf areas were asked to translate 1K utterance-response pairs from the English open-domain dialogues dataset DailyDialog (Li et al., 2017).

### 3.2.12 Grammatical Error Correction

The task of grammatical error correction (GEC) is focused on analyzing written text, automatically pinpointing, and rectifying a variety of grammatical errors as illustrated by a typical instance of grammatical error correction and its manual rectification. In this cluster, we use three GEC datasets: **(1-2) QALB.** We use two datasets extracted from the QALB shared tasks from 2014 (Mohit et al., 2014) and 2015 (Rozovskaya et al., 2015). Both datasets are manually corrected collections of Arabic texts originating from online commentaries on Aljazeera articles written by native Arabic speakers (L1), as well as texts produced by learners of Arabic as a second language (L2). **(3) ZAEBUC.** A corpus that focuses on bilingual writers presented by Habash and Palfreyman (2022). It matches comparable texts in different languages written by the same writer on different occasions. The corpus is enhanced by adding multiple layered annotations, including manually corrected versions of the raw text, allowing us to use it for GEC.

### 3.2.13 Data2Text

The Data2Text (DT) task involves converting structured data like tables as input into descriptive texts without misrepresenting their contents, while sounding natural in writing (i.e., fluently describing this data as output). For the DT task cluster, we use the Arabic subset of the multilingual dataset MD2T proposed by Mille et al. (2020) during the third multilingual surface realization shared task. Table 3 shows examples from each task included in

Dolphin. We now introduce our strong baselines exploiting our benchmark.

### 3.3 Comparative Analysis with ARGEN.

Compared to the previous largest Arabic NLU benchmark, ARGEN (which we list in Table 1), Dolphin (Nagoudi et al., 2022a) exhibits several advantages. Specifically, we observe the following: **Coverage.** Dolphin boasts a significantly larger dataset pool (∼3X larger). In terms of the number of datasets, Dolphin comprises 40 datasets compared to only 13 datasets in ARGEN. Hence, Dolphin offers a total of 27 totally new datasets. **Task clusters.** Dolphin's reach also extends to a wider array of task clusters, encompassing 13 clusters as opposed to ARGEN's *seven* clusters. Dolphin introduces *six* novel tasks: *Arabic text diacritization*, *dialogue response generation*, *data-to-text conversion*, *grammatical error correction*, *text rewriting*, and *question answering*. **Availability**. Dolphin's datasets are drawn exclusively from publicly available sources, while ARGEN involves several non-public datasets such as the machine translation datasets introduced by Zbib et al. (2012) and transliteration presented by Song et al. (2014). As such, Dolphin avoids issues ARGEN suffers from such as challenges with (i) public distribution of the data and (ii) ease of evaluation. **Interactivity**. Dolphin uniquely offers a benchmark leaderboard, a feature absent in ARGEN, providing real-time performance tracking and a dynamic evaluation environment.

## 4 Model Evaluation on Dolphin

In order to establish a conducive environment for meaningful comparisons on Dolphin, we offer a number of strong baselines for both finetuning and *k*-shot settings as described next.

### 4.1 Finetuned Models

For finetuning, we benchmark five different Arabic and multilingual models on Dolphin. These are AraT5 (Nagoudi et al., 2022a), of which we pretrain a new version that we refer to as AraT5$_{v2}$, AraBART (Eddine et al., 2022), mBART (Liu et al., 2020), mT5 (Xue et al., 2020), and mT0 (Muennighoff et al., 2022). More information about these models, including our newly introduced AraT5$_{v2}$, is in Appendix C.

For all models, we finetune on the training data split (Train) for 20 epochs with an early stopping

of 5 epochs, learning-rate of $5e − 5$, batch size of 16, and sequence length of $512$.[9] For all the experiments, we identify the best model on the respective development split (Dev) and blind testing on the test split (Test). We methodically evaluate each task cluster, ultimately reporting a single *Dolphin score* following e.g., Wang et al. (2018) and Elmadany et al. (2023). *Dolphin score* is simply the macro-average of the different scores across task clusters, where each task is weighted equally. Since some of our tasks are reported in metrics where lower numbers are better, we split our metric into *Dolphin$_L$ score* (for tasks where lower ↓ is better [i.e., CER]), and *Dolphin$_H$ score* (for tasks where higher ↑ is better [i.e., BLEU, $F_1$, $M_2$, and ROUGE]). Table 4 presents the results of all pretrained models on each task cluster of Dolphin independently using the relevant metric.

**Discussion.** As Table 4 shows, models dedicated to Arabic outperform multilingual models on tasks where higher is better (in *Dolphin$_H$*). We also note that AraT5$_{v2}$ the new model we build on top of (Nagoudi et al., 2022a), achieves the best Dolphin$_H$ and Dolphin$_L$, at 27.82 and 11.67, respectively. It is followed by AraBART with Dolphin$_H$ of 26.44, where a higher score indicates better performance. Conversely, mT5 achieves a Dolphin$_L$ of 12.42, which is considered better in the opposite scenario. We also note that **AraT5$_{v2}$** achieves the best results in 30 individual tasks out of 50, followed by AraBART and mT0, where each one excels in 11 and 8 individual tasks, respectively.[10]

**Model Computational Costs.** We assess the computational efficiency of the Arabic and multilingual models we finetune. Figure 3 shows for each model the *total time needed for convergence* (under our 20 epochs constraint with a patience of 5) and the conversion epoch. AraBART is the fastest (2.07 hours), with an average of 10.58 epochs to convergence, followed by mT5, AraT5$_{v2}$, mT0, and finally AraT5.

---

[9]Except for GEC, where we use a seq length of $1,024$.

[10]We investigate why AraT5 achieves worst, in spite of being dedicated to Arabic, finding it to perform better with 100 epochs and a patience of 20 as Nagoudi et al. (2022a) report.

| Cluster | Metric | Test Set | mT0 | mT5 | AraBART | AraT5 | AraT5v2 |
|---|---|---|---|---|---|---|---|
| Code-Switching | *Bleu* | *Dz-Fr → Fr* | $10.90^{\pm1.23}$ | $11.92^{\pm0.91}$ | $\mathbf{18.67}^{\pm1.98}$ | $12.23^{\pm2.32}$ | $16.16^{\pm1.68}$ |
| | | *Eg-En → En* | $\mathbf{7.19}^{\pm0.45}$ | $4.38^{\pm1.02}$ | $1.35^{\pm0.65}$ | $2.41^{\pm0.73}$ | $3.22^{\pm0.76}$ |
| | | *Jo-En → En* | $\mathbf{11.37}^{\pm1.11}$ | $8.42^{\pm0.87}$ | $2.0^{\pm0.88}$ | $4.59^{\pm0.32}$ | $6.29^{\pm0.11}$ |
| | | *MA-Fr → Fr* | $11.9^{\pm0.66}$ | $13.63^{\pm0.87}$ | $\mathbf{16.14}^{\pm0.02}$ | $10.87^{\pm0.65}$ | $14.48^{\pm0.32}$ |
| | | *Ps-En → En* | $\mathbf{5.82}^{\pm0.87}$ | $4.84^{\pm0.70}$ | $1.170^{\pm0.91}$ | $2.57^{\pm0.51}$ | $3.67^{\pm0.65}$ |
| | | *Ye-En → En* | $\mathbf{8.59}^{\pm0.07}$ | $6.91^{\pm0.09}$ | $2.8^{\pm0.63}$ | $3.88^{\pm0.76}$ | $5.88^{\pm0.01}$ |
| Data2Text | Bleu | MD2T | $0.22^{\pm0.02}$ | $0.17^{\pm0.06}$ | $0.47^{\pm0.12}$ | $0.04^{\pm0.01}$ | $\mathbf{0.83}^{\pm0.22}$ |
| Diacritization | *CER* | ADT ↓ | $1.58^{\pm0.13}$ | $1.64^{\pm0.11}$ | $23.43^{\pm1.51}$ | $2.58^{\pm0.19}$ | $\mathbf{1.36}^{\pm0.41}$ |
| Dialogue Generation | *Bleu* | AEC | $1.29^{\pm0.21}$ | $1.14^{\pm0.11}$ | $\mathbf{1.71}^{\pm0.03}$ | $1.33^{\pm0.06}$ | $1.41^{\pm0.24}$ |
| | | $DRG_{EGY}$ | $0.05^{\pm0.03}$ | $0.06^{\pm0.04}$ | $\mathbf{0.35}^{\pm0.02}$ | $0.12^{\pm0.03}$ | $0.32^{\pm0.02}$ |
| | | $DRG_{GUL}$ | $\mathbf{1.02}^{\pm0.16}$ | $0.1^{\pm0.07}$ | $0.8^{\pm0.33}$ | $0.29^{\pm0.11}$ | $0.36^{\pm0.12}$ |
| | | $DRG_{LEV}$ | $0.16^{\pm0.11}$ | $0.11^{\pm0.08}$ | $\mathbf{0.57}^{\pm0.20}$ | $0.35^{\pm0.09}$ | $0.48^{\pm0.13}$ |
| GEC | $F_{0.5}\ (M^2)$ | QALB 2014 | $65.86^{\pm0.67}$ | $66.45^{\pm0.22}$ | $68.67^{\pm0.08}$ | $64.92^{\pm0.23}$ | $\mathbf{70.54}^{\pm0.16}$ |
| | | QALB 2015 (L1) | $66.90^{\pm0.92}$ | $66.68^{\pm0.08}$ | $69.31^{\pm1.55}$ | $64.22^{\pm0.82}$ | $\mathbf{70.71}^{\pm0.61}$ |
| | | ZAEBUC | $47.33^{\pm3.34}$ | $46.90^{\pm0.87}$ | $82.08^{\pm7.54}$ | $75.78^{\pm2.43}$ | $\mathbf{84.93}^{\pm4.46}$ |
| Paraphrase | *Bleu* | TAPACO | $15.43^{\pm0.64}$ | $14.89^{\pm0.28}$ | $17.9^{\pm1.06}$ | $15.90^{\pm0.06}$ | $\mathbf{18.69}^{\pm0.26}$ |
| | | APB | $\mathbf{38.36}^{\pm0.14}$ | $24.29^{\pm13.98}$ | $37.66^{\pm1.01}$ | $20.34^{\pm1.82}$ | $30.18^{\pm1.62}$ |
| | | SemEval | $20.49^{\pm0.13}$ | $20.23^{\pm0.03}$ | $24.52^{\pm0.62}$ | $19.33^{\pm0.08}$ | $\mathbf{27.96}^{\pm3.03}$ |
| Question Answering | $F_1$ | $LAREQA_{QA}$ | $\mathbf{63.58}^{\pm0.63}$ | $23.38^{\pm1.12}$ | $45.01^{\pm1.98}$ | $25.45^{\pm2.65}$ | $29.93^{\pm4.73}$ |
| | | $DAWQS_{QA}$ | $2.52^{\pm0.03}$ | $2.82^{\pm0.07}$ | $4.17^{\pm0.30}$ | $0.37^{\pm0.45}$ | $\mathbf{4.98}^{\pm0.08}$ |
| | | $EXAMS_{QA}$ | $42.75^{\pm0.61}$ | $\mathbf{23.24}^{\pm0.55}$ | $22.54^{\pm0.12}$ | $12.69^{\pm0.40}$ | $28.14^{\pm3.80}$ |
| | | $MKQA_{QA}$ | $30.01^{\pm0.41}$ | $32.90^{\pm0.0}$ | $32.42^{\pm0.09}$ | $32.9^{\pm0.0}$ | $\mathbf{33.11}^{\pm0.36}$ |
| | | $LMQA_{QA}$ | $49.17^{\pm0.34}$ | $45.13^{\pm0.35}$ | $47.24^{\pm0.13}$ | $51.95^{\pm0.09}$ | $\mathbf{54.44}^{\pm0.56}$ |
| | | $ARCD_{QA}$ | $53.24^{\pm0.24}$ | $51.63^{\pm1.01}$ | $50.26^{\pm0.99}$ | $58.12^{\pm0.16}$ | $\mathbf{61.38}^{\pm0.97}$ |
| | | $TyDiQA_{QA}$ | $76.31^{\pm0.09}$ | $74.99^{\pm0.23}$ | $73.32^{\pm1.21}$ | $39.55^{\pm1.96}$ | $\mathbf{83.34}^{\pm0.45}$ |
| | | $XQUAD_{QA}$ | $54.55^{\pm0.76}$ | $47.43^{\pm0.91}$ | $47.33^{\pm0.87}$ | $48.71^{\pm0.5}$ | $\mathbf{57.88}^{\pm0.04}$ |
| Question Generation | *Bleu* | $LAREQA_{QG}$ | $9.04^{\pm0.29}$ | $5.5^{\pm2.99}$ | $\mathbf{10.23}^{\pm0.72}$ | $8.65^{\pm0.98}$ | $10.07^{\pm0.56}$ |
| | | Arabic-$SQUAD_{QG}$ | $9.20^{\pm0.07}$ | $9.01^{\pm0.06}$ | $10.10^{\pm0.09}$ | $8.44^{\pm0.11}$ | $\mathbf{10.76}^{\pm0.18}$ |
| | | $MLQA_{QG}$ | $6.04^{\pm0.08}$ | $6.0^{\pm0.38}$ | $7.02^{\pm0.09}$ | $6.12^{\pm0.42}$ | $\mathbf{7.45}^{\pm0.21}$ |
| | | $ARCD_{QG}$ | $17.73^{\pm0.99}$ | $17.62^{\pm2.10}$ | $\mathbf{22.79}^{\pm0.66}$ | $16.8^{\pm1.32}$ | $21.58^{\pm1.55}$ |
| | | $TyDiQA_{QG}$ | $30.22^{\pm0.91}$ | $31.00^{\pm0.97}$ | $\mathbf{33.64}^{\pm0.13}$ | $22.09^{\pm1.85}$ | $\mathbf{33.64}^{\pm0.89}$ |
| | | $XQUAD_{QG}$ | $10.04^{\pm0.01}$ | $9.96^{\pm0.03}$ | $10.27^{\pm0.31}$ | $9.21^{\pm0.09}$ | $\mathbf{10.82}^{\pm0.12}$ |
| Text Rewriting | *Bleu* | APGC | $90.43^{\pm0.14}$ | $90.47^{\pm0.04}$ | $88.93^{\pm0.56}$ | $89.87^{\pm0.07}$ | $\mathbf{91.19}^{\pm0.07}$ |
| | | $DIA2MSA_{EGY}$ | $10.35^{\pm0.58}$ | $10.26^{\pm0.31}$ | $12.57^{\pm0.27}$ | $10.53^{\pm0.08}$ | $\mathbf{14.01}^{\pm0.43}$ |
| Summarization | *RougeL* | XLSum | $21.46^{\pm0.54}$ | $20.64^{\pm0.31}$ | $26.64^{\pm0.04}$ | $22.71^{\pm1.36}$ | $\mathbf{26.88}^{\pm0.02}$ |
| | | CrossSum | $21.0^{\pm0.38}$ | $20.29^{\pm0.01}$ | $25.89^{\pm0.09}$ | $22.14^{\pm1.53}$ | $\mathbf{26.47}^{\pm1.02}$ |
| | | MarSum | $23.0^{\pm0.17}$ | $22.57^{\pm0.21}$ | $\mathbf{26.49}^{\pm0.03}$ | $21.71^{\pm0.39}$ | $25.727^{\pm0.02}$ |
| | | MassiveSum | $25.57^{\pm0.11}$ | $22.88^{\pm0.12}$ | $\mathbf{30.0}^{\pm0.11}$ | $15.89^{\pm0.4}$ | $23.07^{\pm0.33}$ |
| | | ANTCorp | $90.29^{\pm0.11}$ | $88.84^{\pm0.91}$ | $90.0^{\pm0.2}$ | $86.64^{\pm0.22}$ | $\mathbf{91.28}^{\pm0.88}$ |
| Title Generation | *Bleu* | Arabic NTG | $19.03^{\pm0.34}$ | $19.23^{\pm0.01}$ | $\mathbf{22.75}^{\pm0.09}$ | $19.55^{\pm0.16}$ | $22.27^{\pm0.18}$ |
| | | XLSum | $6.50^{\pm0.17}$ | $6.51^{\pm0.11}$ | $8.98^{\pm0.18}$ | $7.44^{\pm0.11}$ | $\mathbf{9.64}^{\pm0.13}$ |
| Transliteration | *CER* | ANTAEC ↓ | $19.21^{\pm0.48}$ | $18.93^{\pm0.30}$ | $18.29^{\pm0.29}$ | $20.74^{\pm0.17}$ | $\mathbf{18.44}^{\pm0.29}$ |
| | *CER* | ATAR ↓ | $16.79^{\pm0.15}$ | $16.68^{\pm0.22}$ | $17.70^{\pm0.05}$ | $36.51^{\pm1.53}$ | $\mathbf{15.20}^{\pm0.32}$ |
| | *Belu* | NETTrans | $55.7^{\pm0.18}$ | $55.02^{\pm0.47}$ | $54.15^{\pm0.75}$ | $51.89^{\pm0.64}$ | $\mathbf{57.41}^{\pm0.93}$ |
| MT | *Bleu* | Darija | $16.95^{\pm1.81}$ | $11.27^{\pm2.54}$ | $16.69^{\pm0.33}$ | $1.29^{\pm0.46}$ | $\mathbf{18.09}^{\pm2.85}$ |
| | | NArabizi | $\mathbf{11.39}^{\pm1.84}$ | $3.37^{\pm0.39}$ | $11.12^{\pm1.20}$ | $6.91^{\pm0.01}$ | $8.98^{\pm1.52}$ |
| | | *En → MSA* | $23.83^{\pm1.04}$ | $23.68^{\pm1.10}$ | $24.13^{\pm0.13}$ | $22.34^{\pm0.13}$ | $\mathbf{28.12}^{\pm0.24}$ |
| | | *Fr → MSA* | $17.28^{\pm0.71}$ | $17.74^{\pm0.08}$ | $17.76^{\pm0.04}$ | $15.73^{\pm0.12}$ | $\mathbf{20.51}^{\pm0.10}$ |
| | | *Es → MSA* | $19.92^{\pm0.7}$ | $20.56^{\pm0.06}$ | $20.38^{\pm0.11}$ | $17.73^{\pm0.20}$ | $\mathbf{21.74}^{\pm0.36}$ |
| | | *Ru → MSA* | $16.93^{\pm0.67}$ | $17.12^{\pm0.18}$ | $3.46^{\pm0.14}$ | $14.10^{\pm0.02}$ | $\mathbf{18.29}^{\pm0.82}$ |
| ***Dolphin_L Score*** | **Avg. ↓ tasks** | | 12.53 | 12.42 | 19.81 | 19.94 | **11.67** |
| ***Dolphin_H Score*** | **Avg. ↑ tasks** | | 26.32 | 23.88 | 26.44 | 22.67 | **27.82** |

Table 4: Average of three runs of finetuned Arabic and multilingual models on Dolphin test. **Dolphin_L Score**: refers to the macro-average scores of tasks where a lower score ↓ is better. **Dolphin_H Score**: refers to the macro-average scores of tasks where a higher score ↑ is better.

| Setting | Few-Shot | | | | | | FFT |
|---|---|---|---|---|---|---|---|
| **Task** | **BLOOMZ** | | | **ChatGPT** | | | **AraT5$_{v2}$** |
| | 0 | 5 | 10 | 0 | 5 | 10 | |
| CST (Jo-en→en) | 11.52 | 11.56 | 11.50 | 36.61 | 38.55 | 40.88 | 5.56 |
| CST (MSA-fr→fr) | 28.41 | 26.75 | 28.61 | 34.61 | 36.45 | 37.95 | 17.49 |
| Diacritization ↓ | 0.51 | 1.62 | 1.42 | 0.11 | 0.05 | 0.06 | **0.02** |
| Dialogue Generation | 0.00 | 0.38 | 0.44 | 0.38 | 0.51 | 0.00 | **0.98** |
| GEC | 26.42 | 28.13 | - | 53.59 | 62.04 | - | **96.63** |
| MT (en→ar) | 8.33 | 12.35 | 10.07 | 20.52 | 23.34 | 23.74 | **26.71** |
| MT (es→ar) | 6.94 | 9.31 | 7.33 | 16.47 | 17.45 | 19.32 | **21.43** |
| MT (fr→ar) | 6.88 | 5.76 | 4.97 | 15.12 | 15.57 | 16.26 | **19.11** |
| MT (ru→ar) | 2.42 | 3.17 | 1.82 | 15.83 | 17.46 | 17.38 | **18.01** |
| Paraphrase | 12.98 | 10.27 | 10.55 | 7.89 | 9.19 | 9.60 | **18.90** |
| Question Answering | 76.04 | 62.08 | 60.49 | 32.98 | 54.14 | 53.67 | **83.16** |
| Question Generation | 28.76 | 18.53 | 18.69 | 14.48 | 20.08 | 18.15 | **34.34** |
| Summarization | 13.56 | 10.74 | 9.63 | 16.88 | 20.40 | 19.58 | **26.96** |
| Text Rewriting | 76.67 | 13.97 | 12.73 | 41.59 | 53.34 | 62.62 | **90.75** |
| Title Generation | 0.99 | 1.20 | 0.62 | 3.24 | 4.62 | 4.54 | **9.30** |
| Transliteration ↓ | 0.59 | 0.42 | 0.42 | 0.27 | 0.24 | 0.23 | **0.20** |

Table 5: *K*-shot results with BLOOMZ and ChatGPT, compared to best finetuned model (AraT5$_{v2}$). We report CER for diacritization and transliteration, ROUGE for summarization, $F_{0.5}$ ($M^2$) for GEC, and $F_1$ for QA. All other tasks reported in BLEU. ↓: lower is better.

## 4.2 Few-Shot Evaluation.

We also carry out *k*-shot evaluations of both BLOOMZ[11] (7.1B) (Muennighoff et al., 2022) and ChatGPT (gpt-3.5-turbo)[12] on 12 different NLG tasks across 16 test sets extracted from Dolphin.[13] To keep the cost manageable, we randomly sample a set of 200 examples from the test set of each task for evaluation. We then evaluate both models under *0-*, *5-*, and *10*-shot settings. For all experiments, we set the temperature to *zero* to generate deterministic and reproducible results. We compare both models' performance to our best fully finetuned model, AraT5$_{v2}$, blind-tested on the same sampled 200 examples.

**Discussion.** Tables 5, shows that ChatGPT outperforms BLOOMZ in all the 16 NLG tasks under *0-*, *5-*, and *10*-shot settings. The only exception is the text rewriting task in the 0-shot setting. It is worth mentioning that AraT5$_{v2}$ outperforms both ChatGPT and BLOOMZ by 14 out of 16. However, ChatGPT (*10*-shot) achieves the highest score in both code-switching tasks, perhaps due to its multilingual pretraining data.

## 5 Conclusion

We presented Dolphin, a large and diverse benchmark for Arabic NLG composed of 40 datasets

---

[11]BLOOMZ is finetuned on multiple tasks in 46 languages, including ∼ 1% Arabic.

[12]We evaluate the version existing on March 1st, 2023.

[13]We only exclude the data-to-text task.

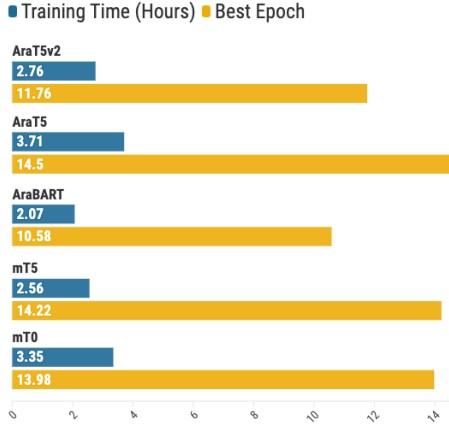

Figure 3: Finetuning time (in hrs) and no. of epoch. We report the average of three runs across all tasks.

that are arranged in 13 tasks. Dolphin is designed to facilitate meaningful comparisons and encourage healthy competition in Arabic. We also provide an interactive leaderboard with a range of useful tools and detailed metadata to help situate future research in a rich context of information sharing. Dolphin datasets are all publicly available, which should facilitate the adoption and further development of the benchmark. In the future, we intend to build on top of Dolphin by extending it to more tasks and Arabic varieties.

## 6 Limitations

In spite of the diversity, wide-coverage, high-quality datasets, accessibility, and challenging nature of Dolphin, it is not without limitations. In particular, we identify the following limitations.

1. **Coverage of Arabic Varieties.** While we make efforts to incorporate tasks from all Arabic varieties, it is important to note that there is a lack of available downstream datasets from countries such as Djibouti, Mauritania, and Yemen. Consequently, these varieties are not currently included in Dolphin. We hope that the community will develop resources representing all Arab countries, including these, across the various tasks. We also hope that future versions of our benchmark will have extended dialectal coverage in ways that enhance its representation of the Arabic language and help foster technological inclusion.

2. **Machine-Translated Datasets.** Dolphin includes two machine-translated

data, AEC (Naous et al., 2021) and Ara-Para (Nagoudi et al., 2022a)). While these datasets increase task coverage in Dolphin, the MT process may inadvertently introduce some biases. For example, MT can result in a narrow representation of language patterns and structures, leading to a limited understanding of the complexities and nuances of different languages. Additionally, benchmark datasets may not adequately capture the wide range of domains, genres, and styles that exist in real-world translation scenarios. This can limit the generalizability of models trained on such data, as they may struggle to handle unfamiliar or specialized content. We hope that future versions of Dolphin will involve real-world data that further complement (or even substitute) these translated datasets.

3. **Automated Evaluation.** Although all NLP depends heavily on automated evaluation to speed up model development, automated methods have their limitations, especially for some tasks. That is, in addition to automated evaluation, some tasks may need human evaluation. In particular, we believe human evaluation can play a crucial role in NLG tasks such as open-domain dialogue generation. For example, it can capture the nuanced aspects of dialogue quality, such as coherence, relevance, and appropriateness. In addition, human evaluation can allow for a comprehensive assessment of the generated dialogues, taking into account contextual understanding, fluency, and overall user experience. This feedback is invaluable in refining and improving dialogue generation models, ensuring that they meet the high standards of human-like conversation.

## 7 Ethics Statement

**Data Collection and Release.** Dolphin is based on publicly available datasets that would not be possible without the hard work of a large number of researchers over the years. We are grateful for these efforts invested by pioneer colleagues. One downside of benchmarking could be that the original authors of the different datasets are not sufficiently acknowledged. In our work, we make sure that all publications of resources we use are properly cited, both by referencing these in this paper (Section 3) and highlighting them in our GitHub and leaderboard website.

1. **Data Privacy.** Regarding data involved in Dolphin, we develop the benchmark using publicly available data. For this reason, we do not have significant privacy concerns. In addition, the new datasets we develop and release for code-switched machine translation have undergone manual inspection to ensure there is no unintended leak of privacy information in any of the samples.

2. **Intended Use.** We believe our work will spur further research on studying language models on Arabic NLG benchmark. We create a publicly available leaderboard and benchmark several multilingual and Arabic-dedicated SOTA models on Dolphin. The benchmark will facilitate a unified evaluation and pave the way for a healthy competition that could push SoTA on Arabic language generation.

3. **Potential Misuse and Bias.** The datasets we collect to create Dolphin may contain potential harmful contents. Additionally, the models we evaluate might be exposed to bias and as a result may generate unintended contents. Therefore, we recommend that these datasets and models not be used in applications without careful prior consideration of potential misuse and bias.

## Acknowledgements

We gratefully acknowledge support from Canada Research Chairs (CRC), the Natural Sciences and Engineering Research Council of Canada (NSERC; RGPIN-2018-04267), the Social Sciences and Humanities Research Council of Canada (SSHRC; 435-2018-0576; 895-2020-1004; 895-2021-1008), Canadian Foundation for Innovation (CFI; 37771), Digital Research Alliance of Canada,[14] UBC ARC-Sockeye.[15] We thank the Google TFRC program for providing us with free TPU access.[16]

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

# Appendices

We organize our appendices as follows:

**Sections list**:

**Tables and Figures List**:

## A NLG Benchmarks

Existing NLG benchmarks can be classified into three distinct categories: *Arabic-specific*, *X-specific* (where X refers to languages other than Arabic, such as English, Chinese, and others), and *multilingual* benchmarks. In this section, we shall provide a brief overview of each category, highlighting their respective characteristics and scope. We will highlight aspects such as the target language, dataset size, and the breadth of tasks covered. This analysis is summarized in Table 1 and Figure 2. The current NLG benchmarks can be divided into three main groups: benchmarks that focus on Arabic, benchmarks that focus on languages other than Arabic (X-specific), and benchmarks that cover multiple languages. In this section, we will give a brief summary of each category, emphasizing their unique features and scope. We will discuss factors like the target language, dataset size, and the range of tasks included.

### A.1 Arabic Benchmarks

**AraBench.** AraBench is an evaluation benchmark for dialectal Arabic to English machine translation (MT) introduced by (Sajjad et al., 2020). It consists of five publicly available datasets: Arabic-Dialect/English Parallel Text (APT) (Zbib et al., 2012), Multi-dialectal Parallel Corpus of Arabic (MDC) (Bouamor et al., 2014), MADAR Corpus (Bouamor et al., 2018), Qatari-English speech corpus (Elmahdy et al., 2014), and the English Bible translated into MSA, Tunisian, and Morocco.[17]

***AraOPUS-20.*** This is an MT benchmark proposed by Nagoudi et al. (2022b). It consists of parallel bitext between Arabic and 20 languages extracted from the OPUS publicly available corpora (Tiedemann, 2012). The languages paired with Arabic include high-resource languages such as *English, French*, and *Spanish* and low-resource ones such as *Cebuano*,[18] *Tamashek*,[19] and *Yoruba*.[20]

**ARGEN.** The **AR**abic natural language **GEN**eration (**ARGEN**) benchmark was introduced by Nagoudi et al. (2022a). It is composed of 19 datasets and covers the seven tasks: machine translation, code-switched text translation, summarization, news title generation, question generation, paraphrasing, and transliteration.

### A.2 X-Specific Benchmarks

**GLGE.** The **G**eneral **L**anguage **G**eneration **E**valuation(GLGE) by Liu et al. (2021) is a multi-task benchmark for evaluating the generalization capabilities of NLG in the English language. GLGE has eight English language generation datasets, covering four NLG tasks: data-to-text, dialog, table-to-text, and summarization.

**BanglaNLG.** BanglaNLG is a benchmark designed for Bangala Bhattacharjee et al. (2023) comprising seven datasets across six NLG tasks: machine translation, text summarization, question answering, dialogue generation, headline generation, and cross-lingual summarization.

**CUGE.** The **C**hinese Language **U**nderstanding **G**eneration **E**valuation **B**enchmark Yao et al. (2021) covers both language understanding and generation. The language generation collection contains nine datasets across eight tasks. The tasks are open-domain question answering, document retrieval, summarization, data-to-text, knowledge-driven conversation, machine translation, cross-lingual text summarization, and mathematical computation. The benchmark also covers the tasks of grammatical error correction and reverse dictionary generation, but treats these under the NLU component.

---

[17]The United Bible Societies https://www.bible.com

[18]Language spoken in the southern Philippines

[19]*Tamashek* is a variety of Tuareg, a Berber macro-language widely spoken by nomadic tribes across North Africa countries.

[20]Yoruba is a language spoken in West Africa, primarily in Southwestern Nigeria.

**Bahasa Indonesia.** The Bahasa Indonesia language has over 200M active speakers, yet it is still considered a low-resource language. To overcome this problem, (Guntara et al., 2020) introduced a machine translation benchmark with 14 datasets across four domains: news, religion, conversation, and general.

**PhoMT.** Doan et al. (2021) introduces a new Vietnamese-English parallel dataset that is larger and of higher quality than the existing benchmark corpus. The authors conduct experiments to evaluate various translation models on the new dataset and find that the best performance is achieved by fine-tuning the pre-trained sequence-to-sequence denoising auto-encoder mBART.

**LOT.** The **LO**ng **T**ext understanding and generation benchmark targets Chinese long text modeling in a story-centric manner Guan et al. (2022). LOT combines two comprehension tasks and two-generation tasks. The two generation tasks are commonsense reasoning and discourse structure.

### A.3 Multi-Lingual NLG Benchmarks

**IndoNLG.** IndoNLG covers three low resources languages widely spoken in Indonesia: Indonesian, Javanese, and Sundanese Cahyawijaya et al. (2021). It consists of ten distinct datasets, encompassing four tasks. These are summarization, question answering, chit-chat, and machine translation.

**CLSE.** The **C**orpus of **L**inguistically **S**ignificant **E**ntities Chuklin et al. (2022) is a multilingual named entities corpus that covers 34 languages, 74 semantic classes, and 222 distinguishable linguistic signatures. The authors also developed an expanded version of the Schema-Guided Dialog Dataset (SG-CLSE) to illustrate one of the potential uses of CLSE in three languages: French, Marathi, and Russian.

**GEM$_{v1}$.** The **G**eneration **E**valuation and **M**etrics benchmark (Gehrmann et al., 2021) is a multilingual benchmark environment for NLG. GEM features 18 languages across 13 datasets spanning five NLG tasks: data-to-text, dialog response generation, reasoning, summarization, and simplification.[21]

**GEM$_{v2}$.** Gehrmann et al. (2022) propose a second version, GEM$_{v2}$, styled after GEM$_{v1}$ with a new set of datasets and more challenging tasks. This new version supports 40 documented datasets in 51 languages. It introduces a modular infrastructure

for datasets and models, with an online evaluation process that collects model outputs and computes metrics for all datasets. GEM$_{v2}$ is built around nine NLG tasks data-to-text, dialog response generation, paraphrasing, generative question answering, question generation, reasoning, slide generation, simplification, and summarization.

**IndicNLG.** The first benchmark for Indic languages Kumar et al. (2022) covers 11 Indic languages belonging to two language families: Indo-Aryan and Dravidian. IndicNLG involves the five following tasks: biography generation, news headline generation, sentence summarization, paraphrase generation, and question generation.

**MTG.** Chen et al. (2022) introduce the **M**ultilingual **T**ext **G**eneration to promote knowledge transfer and cross-lingual generation between arbitrary language pairs. MTG contains 400K of humanly annotated data samples in five languages, covering four generation tasks. These are story generation, question generation, title generation, and text summarization.

## B Dolphin Tasks

## C Arabic and Multilingual S2S LLMs

In this section, we list the Arabic and multilingual sequence-to-sequence (S2S) pretrained LMs we finetune on Dolphin.

**AraT5.** (Nagoudi et al., 2022a) is an adaptation of the T5 model specifically designed for the Arabic language. It is pre-trained on a large (248GB of Arabic text) diverse (MSA and Arabic dialects) dataset to effectively handle different Arabic tasks. In addition to Arabic, AraT5's vocabulary covers 11 other languages. In this work, we evaluate a new in-house version of AraT5 dubbed AraT5$_{v2}$.

**AraT5$_{v2}$.** Our analysis shows that AraT5 requires a large number of epochs to converge, making it an expensive model. For this reason, we pretrain a new version of the model from scratch exploiting a larger ($\sim$ 400GB) and more diverse pretraining dataset than used by (Nagoudi et al., 2022a). As we show in our results, the new model converges faster than AraT5 and achieves better results under our cap of 20 epochs for finetuning across all models.

**AraBART.** (Eddine et al., 2022) is a model based on the encoder-decoder BART base architecture (Lewis et al., 2020), featuring six encoder and 6 decoder layers. It is pretrained on the same corpus as AraBERT (Antoun et al., 2020), with reversed preprocessing for more natural text gener-

---

[21]Two of the datasets do not include English at all.

| Task Cluster | Task | Test Set | Source | Train[*] | Dev[†] | Test[‡] |
|---|---|---|---|---|---|---|
| **MT** | $X \rightarrow MSA$ | $De \rightarrow Ar$ | Eisele and Chen (2010)[*] | 50K | 4K | 4K |
| | | $En \rightarrow Ar$ | | 50K | 4K | 4K |
| | | $Fr \rightarrow Ar$ | Ziemski et al. (2016)[†‡] | 50K | 4K | 4K |
| | | $Ru \rightarrow Ar$ | | 50K | 4K | 4K |
| | $Arabizi \rightarrow X$ | $Dz \rightarrow Fr$ | Seddah et al. (2020) | 1.1K | 144 | 146 |
| | | $Ma \rightarrow En$ | Outchakoucht and Es-Samaali (2021) | 8K | 2K | 2K |
| | $DA \rightarrow En$ | $Eg \rightarrow En$ | | | 200 | 800 |
| | | $Jo \rightarrow En$ | | | 200 | 800 |
| | | $Ps \rightarrow En$ | Nagoudi et al. (2022b)[*] | 50K | 200 | 800 |
| | | $Sy \rightarrow En$ | Bouamor et al. (2014)[†‡] | | 200 | 800 |
| | | $Tn \rightarrow En$ | | | 200 | 800 |
| **Code-Switching** | $DA\text{-}X \rightarrow X$ | $Dz\text{-}Fr \rightarrow Fr$ | | | 50 | 250 |
| | | $Ma\text{-}Fr \rightarrow Fr$ | | | 50 | 250 |
| | | $Tn\text{-}Fr \rightarrow Fr$ | Nagoudi et al. (2022b)[*] | 50K | 50 | 250 |
| | | $Eg\text{-}En \rightarrow En$ | *Our work*[†‡] | | 50 | 250 |
| | | $Jo\text{-}En \rightarrow En$ | | | 50 | 250 |
| | | $Ps\text{-}Fr \rightarrow En$ | | | 50 | 250 |
| **Summarization** | $MSA \rightarrow MSA$ | $ANT Corpus$ | Chouigui et al. (2021) | 25.2K | 3.1K | 3.1K |
| | | $CrossSum$ | Bhattacharjee et al. (2021) | 37.3K | 4.6K | 4.7K |
| | | $MassiveSum$ | Varab and Schluter (2021) | 4.6K | 459 | 1.3K |
| | | $XLSum$ | Hasan et al. (2021) | 37.5K | 4.7K | 4.7K |
| | $DA \rightarrow DA$ | $MarSum$ | Gaanoun et al. (2022) | 16K | 1.7K | 1.9K |
| **Title Generation** | $MSA \rightarrow MSA$ | $Arabic NTG$ | Nagoudi et al. (2022a) | 93.3.5K | 11.6K | 11.6K |
| | | $XLSum$ | Hasan et al. (2021) | 37.5K | 4.7K | 4.7K |
| **QA/QG** | $MSA \rightarrow MSA$ | $ARCD$ | Mozannar et al. (2019)[†‡] | 49.9K | 693 | 702 |
| | | $MLQA$ | Lewis et al. (2019)[†‡] | 49.9K | 517 | 5.3K |
| | | $XQuAD$ | Artetxe et al. (2020)[‡] | 49.9K | 5.08K | 1.1K |
| | | $TyDiQA$ | Artetxe et al. (2020)[*‡] | 49.9K | 5.08K | 921 |
| | | $LAReQA$ | Roy et al. (2020) | 851 | 119 | 220 |
| | | $DAWQAS$ | Ismail and Nabhan Homsi (2018) | 2.2K | 318 | 645 |
| | | $EXAMS$ | Hardalov et al. (2020) | 7.9K | 2.6K | 13.5K |
| **Transliteration** | $Arabizi \rightarrow MSA$ | $ANETAC$ | Ameur et al. (2019) | 75.9K | 1K | 3K |
| | | $ATAR$ | Talafha et al. (2021) | 17.2K | 2.1K | 2.1K |
| | | $NETTrans.$ | Merhav and Ash (2018) | 116K | 14.5K | 14.5K |
| **Text Rewriting** | $DA \rightarrow MSA$ | $Egy \rightarrow MSA$ | | 3.8K | 551 | 1.1K |
| | | $Mag \rightarrow MSA$ | Mubarak (2018) | 3.4K | 491 | 996 |
| | | $Lev \rightarrow MSA$ | | 4.2K | 594 | 1.2K |
| | | $Gul \rightarrow MSA$ | | 4.2K | 594 | 1.2K |
| | $MSA \rightarrow MSA$ | $APGC$ | Alhafni et al. (2022) | 40.4K | 4.7K | 11.3K |
| **Diacritization** | CA $\rightarrow$ CA | $ATD$ | Fadel et al. (2019) | 50K | 2.5K | 2.5K |
| **Data2Text** | Table $\rightarrow$ MSA | $MD2T$ | Mille et al. (2020) | 6K | 900 | 680 |
| **Dialogue Generation** | MSA$\rightarrow$ MSA | $AEC$ | Naous et al. (2020) | 32.9K | 1.8K | 1.8K |
| | $DA \rightarrow DA$ | $Egy \rightarrow Egy$ | | 2.1K | 297 | 600 |
| | | $Lev \rightarrow Lev$ | Naous et al. (2023) | 2.1K | 297 | 600 |
| | | $Gul \rightarrow Gul$ | | 2.1K | 297 | 600 |
| **GEC** | MSA $\rightarrow$ MSA | $QALB 2014$ | Mohit et al. (2014) | 19.4K | 1K | 968 |
| | | $QALB 2015$ | Rozovskaya et al. (2015) | 310 | 154 | 158 |
| | | $ZAEBUC$ | Habash and Palfreyman (2022) | 27K | 3.3K | 3.3K |
| **Paraphrase** | MSA $\rightarrow$ MSA | $ASEP$ | Cer et al. (2017) | 116.4K | 6.1K | 600 |
| | | $APB$ | Alian et al. (2019) | 808 | 202 | 101 |
| | | $TaPaCo$ | Scherrer (2020) | 2.1K | 299 | 605 |

Table B.2: Statistics of our *Dolphin* benchmark across the different task clusters. For the QA task, we use the Arabic machine translated SQuAD (AR-XTREME$_{train}$) from Hu et al. (2020) as Train for ARCD, MLQA, and XQuAD. We also use AR-XTREME$_{dev}$ as Dev for XQuAD and TyiQA, respectively. For ASEP (Cer et al., 2017) test set in the summarization task, we use AraPara$_{Train}$ and AraPara$_{Dev}$.

ation. AraBART is designed for various NLP tasks, demonstrating robust performance across different tasks in the Arabic language.

**mBART.** A multilingual encoder-decoder model proposed by Liu et al. (2020). mBART is pretrained by denoising full texts in 50 languages, including Arabic. Then, it is finetuned on parallel MT data contains a total of 230M parallel sentences under three settings: individually toward English and vice versa (i.e., *many-to-English*, and *English-to-many*), or between multiple languages simultaneously (many-to-many).

**mT5.** (Xue et al., 2020) is a multilingual variant of the of **T**ext-**t**o-**T**ext **T**ransfer **T**ransformer model (T5) (Raffel et al., 2019) that covers 101 languages. It is pretrained on a new Common Crawl-based dataset ($\sim$ 26.76TB), designed to achieve SOTA performance on a variety of multilingual NLP tasks such as question answering, document summarization, and MT.

**mT0.** (Muennighoff et al., 2022) is a group of sequence-to-sequence models ranging in size between 300M to 13B parameters trained to investigate the cross-lingual generalization through multitask finetuning. The models are finetuned from preexisting mT5 (Xue et al., 2020) multilingual language models using a cross-lingual task dataset called xP3. mT0 models can execute human instructions in many languages without any prior training.

## D Leaderboard

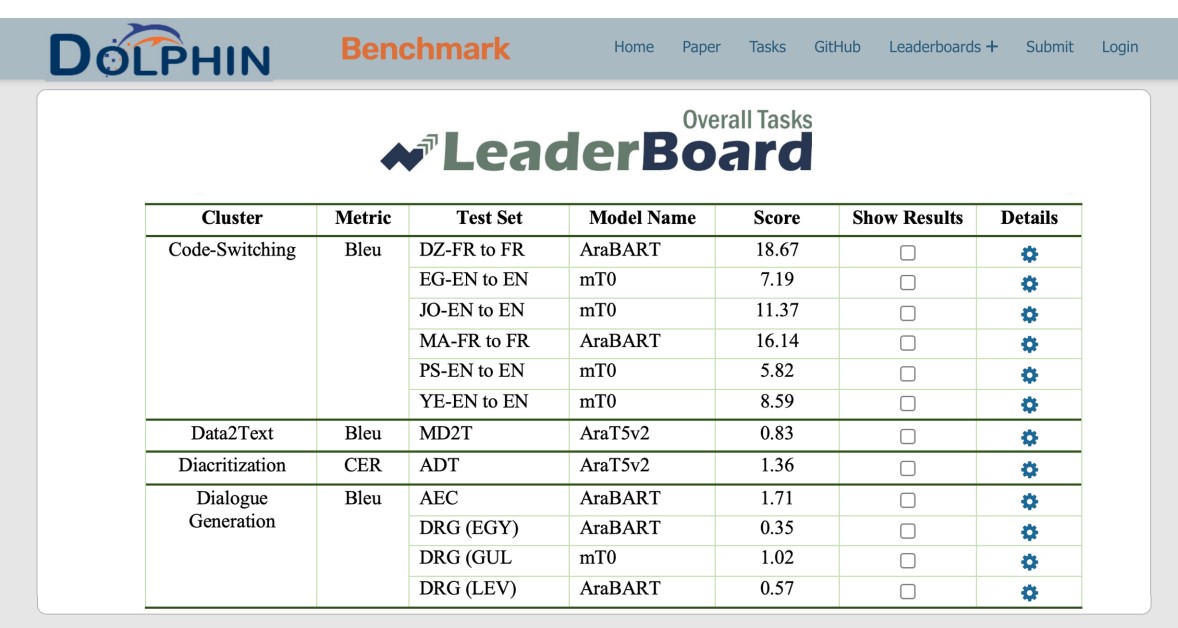

Figure D.1: Dolphin main leaderboard allows showing detailed scores by all models for a given task.