# OpenReview forum: "Dolphin: A Challenging and Diverse Benchmark for Arabic NLG"
_EMNLP/2023/Conference — EMNLP 2023 Findings_

### Official Review · Reviewer_cCB7 · 2023-08-04

**Soundness:** 3

**Excitement:**

3: Ambivalent: It has merits (e.g., it reports state-of-the-art results, the idea is nice), but there are key weaknesses (e.g., it describes incremental work), and it can significantly benefit from another round of revision. However, I won't object to accepting it if my co-reviewers champion it.

**Paper Topic And Main Contributions:**

The authors of this paper introduced a benchmark for Arabic natural language generation (NLG) consisting of 13 tasks curated from 40 datasets. The authors evaluate their benchmark on multiple multilingual pre-trained language models for both finetuning and k-shot settings.

**Reasons To Accept:**

The paper's organization and presentation are commendable overall.

The work exhibits many experiment results of multilingual pre-trained language models on their novel benchmark.

**Reasons To Reject:**

Similar to ARGEN, the Dolphin benchmark just collected 40 datasets and combined them into a benchmark. The design principles used in the papers are also mentioned in the ARGEN benchmark paper.

The discussion of model evaluation on Dolphin does not consider the signatures of the Dolphin benchmark, including diverse and inclusive, or any specified tasks.

The "Model Computational Costs" part of the Discussion is meaningless.




**Reproducibility:**

3: Could reproduce the results with some difficulty. The settings of parameters are underspecified or subjectively determined; the training/evaluation data are not widely available.

**Reviewer Confidence:**

5: Positive that my evaluation is correct. I read the paper very carefully and I am very familiar with related work.

---

> ### Author Rebuttal · Authors · 2023-08-29
>
> Thanks so much for your review.
>
> **Reason to reject #1:** *“similar to ARGEN”*
>
> **Answer:** In the context of comparing Dolphin with ARGEN, as highlighted within lines 119-123 and Table 1, several significant distinctions become apparent. Specifically, we observe the following:
>
> - Dolphin boasts a significantly larger dataset pool (~ 3X larger). Precisely, Dolphin comprises 40 datasets compared to only 13 datasets in ARGEN. Hence, Dolphin offers a total of 27 totally new datasets.
> - Dolphin's reach also extends to a wider array of task clusters, encompassing 13 clusters as opposed to ARGEN's seven clusters. Dolphin introduces six novel tasks: Arabic text diacritization, dialogue response generation, data-to-text conversion, grammatical error correction, text rewriting, and question answering.
> - Dolphin's datasets are drawn exclusively from publicly available sources, while ARGEN involves several non-public datasets such as Zbib et al. [2012] and Song et al. [2014]. As such, Dolphin avoids issues ARGEN suffers from such as challenges with (i) public distribution of the data and (ii) ease of evaluation.
> - Dolphin uniquely offers a benchmark leaderboard, a feature absent in ARGEN, providing real-time performance tracking and a dynamic evaluation environment.
>
> **Reason to reject #2:** *“the Dolphin benchmark just collected 40 datasets and combined them into a benchmark.”*
>
> **Answer:** Compilation of existing datasets to establish a benchmark is an important and established tradition in NLP that is shared, for example by other benchmarking efforts such as GLUE (Wang et al., 2018), SuperGLUE (Wang et al., 2019), GEMv1 (Gehrmann, 2021), GEMv2 (Gehrmann, 2022), MTG [Chen et al.,2022], etc. However, we would like to emphasize that our work involves much more than simple collection of existing datasets in that (1) we meticulously curate, select, standardize formatting and experimental conditions of a large number of datasets; (2) we thematically categorize the datasets across several task clusters, (3) we characterize the extent to which the datasets represent a diverse array of challenging NLG tasks across different Arabic varieties (CA, dialects, and MSA); (4) We create six new human-written (natural) code-switched parallel test datasets that we incorporate into the benchmark; (5) we exploit the datasets to compare different Arabic language models suited for NLG; and (6) we create a highly modular and interactive leaderboard to allow for measurement of progress on Arabic and multilingual NLG.
>
> **Reason to reject #3:** *“ The design principles used in the papers are also mentioned in the ARGEN benchmark paper.”*
>
> **Answer:** The ARGEN benchmark paper does not discuss any design principles. Again, our work is significantly different from the work introducing the ARGEN benchmark and Dolphin itself is completely different from ARGEN as we emphasize in our response to the first point above. For example, ARGEN does not entertain dataset accessibility (it has private datasets) nor ease of meaningful comparison via a public leaderboard. In addition, it does not claim representativeness of Arabic varieties (e.g., it has no coverage of CA).
>
> **Reason to reject #4:** *“The discussion of model evaluation on Dolphin does not consider the signatures of the Dolphin benchmark, including diverse and inclusive, or any specified tasks. The "Model Computational Costs" part of the Discussion is meaningless. “*
>
> **Answer:** Our discussion of Dolphin does cover it being diverse and inclusive (Figure 1, lines 176-194, and Table C.1). However, we will strengthen and deepen this discussion. We will also offer a stronger and explicit comparison between performance of each of the models on the different varieties (CA, DA, MSA).
>
> We provide the section on model computational cost since we believe it is useful for practitioners who have limited compute resources. It can be helpful for choosing which model to use under which compute budget. We will explain this motivation along with a thorough analysis and comparison of the computational expenses of the different models.

---

### Official Review · Reviewer_5EDb · 2023-08-05

**Typos Grammar Style And Presentation Improvements:** 1) In the caption of Figure 1, I beli…
**Soundness:** 3

**Excitement:**

4: Strong: This paper deepens the understanding of some phenomenon or lowers the barriers to an existing research direction.

**Paper Topic And Main Contributions:**

The paper introduces a comprehensive benchmark for Arabic NLG comprising 40 datasets over 13 tasks. To create that they mostly rely on already publicly available datasets and they additionally create datasets for context switching and dialogue response generation. A leaderboard and an investigation of baseline performances on the benchmark are also included. The authors promote the benchmark as an important resource that can help boost research in Arabic NLG.

The introduced leaderboard comes with an aggregate metric called Dolphin score, which is the macro-average of the scores across all tasks. This manner of having a single score is beneficial for leaderboarding. Additionally, more granular performance reports are available.

**Questions For The Authors:**

1) Are there any major already available Arabic NLG datasets that you did not include? If so, why?

**Reasons To Accept:**

1) I do believe this is a useful resource for Arabic NLG. I support efforts like this that provide standard and structured benchmarks for Languages other than English.
2) The benchmark is designed in a way to honor easy public access, diversity, and referencing the original creators of a datasets if it's included from a prior work.
3) This will help with the reproducibility of research in Arabic NLG.

**Reasons To Reject:**

N/A

**Reproducibility:**

4: Could mostly reproduce the results, but there may be some variation because of sample variance or minor variations in their interpretation of the protocol or method.

**Reviewer Confidence:**

4: Quite sure. I tried to check the important points carefully. It's unlikely, though conceivable, that I missed something that should affect my ratings.

---

> ### Author Rebuttal · Authors · 2023-08-29
>
> Thanks so much for your review.
>
> **Reason to reject:** *"The introduced leaderboard comes with an aggregate metric called Dolphin score, which is the macro-average of the scores across all tasks. While this is a way to do aggregate reporting, I wonder if it can conceal failures in the case of specific tasks. I understand this manner of having a single score is beneficial for leader boarding. But I'd also advocate for more granular performance reports. To be fair, this is something that's indeed supported by their leaderboard. I'm just mentioning it to emphasize it."*
>
> **Answer:** Thank you for the review. We appreciate your emphasis about the importance of including granular scores that allow revealing model performance on specific tasks. As you mention, our leaderboard does support such a level of granularity. That is, the leaderboard offers a score for each specific task (at the level of each individual dataset). This is in addition to the benchmark-level scores. As illustrated in Table 2 and Figure F.1 (Appendix F), our leaderboard introduces a suite of comprehensive options for the assessment of Arabic language models. These options encompass not only comparisons between models across individual tasks, but also gauge performance on task clusters focused on achieving lower scores (Dolphin_L Score), as well as task clusters where higher scores signify heightened performance (Dolphin_H Score). Since you clearly pointed out that the leaderboard has this level of granularity in your important statement reading “[t]o be fair, this is something that's indeed supported by their leaderboard. I'm just mentioning it to emphasize it”, we will appreciate you considering removing this as a reason to reject the paper. We very much appreciate you striving to be fair.
>
> **Q1:** *Are there any major already available Arabic NLG datasets that you did not include? If so, why?*
>
> **Answer:** To the best of our knowledge, there are no major publicly available datasets for Arabic NLG that we did not include. In fact, we have undertaken extensive searches to trace any publicly available dataset for inclusion in our benchmark. That is, our benchmark does not ignore any publicly available dataset we could locate even if such a dataset was not “major”. In other words, we strived to make Dolphin both exhaustive and inclusive. In addition, our intention is to enhance the benchmark and our interactive leaderboard in the future by adding datasets that will be newly created.
>
> **Typos:** We will correct the typos mentioned. Thank you!

---

### Official Review · Reviewer_rLjw · 2023-08-12

**Paper Topic And Main Contributions:** 1) The paper introduces a novel Arabi…
**Soundness:** 4

**Excitement:**

4: Strong: This paper deepens the understanding of some phenomenon or lowers the barriers to an existing research direction.

**Reasons To Accept:**

> the paper introduces a new, comprehensive, and well-constructed benchmark with a clear comparison to existing benchmarks
> the paper provides baselines, analysis, and evaluation framework that addresses and highlights strengths and weaknesses in existing works across different generative tasks which is very beneficial to researchers in this field

This is a high-impact paper well-suited to a high-impact conference.



**Reasons To Reject:**

No reasons to reject.

**Reproducibility:**

5: Could easily reproduce the results.

**Reviewer Confidence:**

4: Quite sure. I tried to check the important points carefully. It's unlikely, though conceivable, that I missed something that should affect my ratings.

**Typos Grammar Style And Presentation Improvements:**

(2, 80) Abbreviation MT is first introduced without definition.
(2, 96) Space between dolphin and design is missing
(3) It is better to have Table 1 sorted by date

---

> ### Author Rebuttal · Authors · 2023-08-29
>
> Thanks so much for your review.

---

### Meta-Review · Area_Chair_r8UW · 2023-09-15

**Recommendation:** 4

**Metareview:**

The paper introduces a comprehensive benchmark for Arabic NLG comprising 40 datasets over 13 tasks. To create that they mostly rely on already publicly available datasets and they additionally create datasets for context switching and dialogue response generation. A leaderboard and an investigation of baseline performances on the benchmark are also included. The authors promote the benchmark as an important resource that can help boost research in Arabic NLG.

Reasons To Accept:
- Introduction of a new, comprehensive, and well-constructed benchmark with a clear comparison to existing benchmarks
- The paper provides baselines, analysis, and evaluation framework that addresses and highlights strengths and weaknesses in existing works across different generative tasks which is very beneficial to researchers in this field
- This is a high-impact paper well-suited to a high-impact conference.
-Useful resource for Arabic NLG.
- Public access and diversity
- Help with the reproducibility of research in Arabic NLG.

Reasons To Reject:
- Similar to ARGEN. The design principles used in the papers are also mentioned in the ARGEN benchmark paper.

In my opinion the reason to reject is not fair, and the answer written by the authors is explicit:
"In the context of comparing Dolphin with ARGEN, as highlighted within lines 119-123 and Table 1, several significant distinctions become apparent. Specifically, we observe the following:
- Dolphin boasts a significantly larger dataset pool (~ 3X larger). Precisely, Dolphin comprises 40 datasets compared to only 13 datasets in ARGEN. Hence, Dolphin offers a total of 27 totally new datasets.
- Dolphin's reach also extends to a wider array of task clusters, encompassing 13 clusters as opposed to ARGEN's seven clusters. Dolphin introduces six novel tasks: Arabic text diacritization, dialogue response generation, data-to-text conversion, grammatical error correction, text rewriting, and question answering.
- Dolphin's datasets are drawn exclusively from publicly available sources, while ARGEN involves several non-public datasets such as Zbib et al. [2012] and Song et al. [2014]. As such, Dolphin avoids issues ARGEN suffers from such as challenges with (i) public distribution of the data and (ii) ease of evaluation.
- Dolphin uniquely offers a benchmark leaderboard, a feature absent in ARGEN, providing real-time performance tracking and a dynamic evaluation environment."

---

### Decision · Program_Chairs · 2023-10-07

**Decision:**

Accept-Findings

**Comment:**

The paper introduces a comprehensive benchmark for Arabic NLG comprising 40 datasets over 13 tasks. To create that they mostly rely on already publicly available datasets and they additionally create datasets for context switching and dialogue response generation. A leaderboard and an investigation of baseline performances on the benchmark are also included. The authors promote the benchmark as an important resource that can help boost research in Arabic NLG.

Reasons To Accept:
- Introduction of a new, comprehensive, and well-constructed benchmark with a clear comparison to existing benchmarks
- The paper provides baselines, analysis, and evaluation framework that addresses and highlights strengths and weaknesses in existing works across different generative tasks which is very beneficial to researchers in this field
- This is a high-impact paper well-suited to a high-impact conference.
-Useful resource for Arabic NLG.
- Public access and diversity
- Help with the reproducibility of research in Arabic NLG.

Reasons To Reject:
- Similar to ARGEN. The design principles used in the papers are also mentioned in the ARGEN benchmark paper.

In my opinion the reason to reject is not fair, and the answer written by the authors is explicit:
"In the context of comparing Dolphin with ARGEN, as highlighted within lines 119-123 and Table 1, several significant distinctions become apparent. Specifically, we observe the following:
- Dolphin boasts a significantly larger dataset pool (~ 3X larger). Precisely, Dolphin comprises 40 datasets compared to only 13 datasets in ARGEN. Hence, Dolphin offers a total of 27 totally new datasets.
- Dolphin's reach also extends to a wider array of task clusters, encompassing 13 clusters as opposed to ARGEN's seven clusters. Dolphin introduces six novel tasks: Arabic text diacritization, dialogue response generation, data-to-text conversion, grammatical error correction, text rewriting, and question answering.
- Dolphin's datasets are drawn exclusively from publicly available sources, while ARGEN involves several non-public datasets such as Zbib et al. [2012] and Song et al. [2014]. As such, Dolphin avoids issues ARGEN suffers from such as challenges with (i) public distribution of the data and (ii) ease of evaluation.
- Dolphin uniquely offers a benchmark leaderboard, a feature absent in ARGEN, providing real-time performance tracking and a dynamic evaluation environment."